# Woven EndoBridge in Wide-Neck Bifurcation Aneurysms: Digital Subtraction Angiography at 3-Year Follow-Up

**DOI:** 10.3390/jcm11102879

**Published:** 2022-05-19

**Authors:** Carmelo Stanca, Serena Carriero, Davide Negroni, Marco Spinetta, Carolina Coda, Pierpaolo Biondetti, Giuseppe Guzzardi

**Affiliations:** 1Maggiore della Carità Hospital of Novara, 28100 Novara, Italy; carmelostanca@gmail.com (C.S.); dvdngr@gmail.com (D.N.); marcospinetta90@gmail.com (M.S.); carolinacoda94@gmail.com (C.C.); 2Radiodiagnostic and Interventional Radiology Department, Università degli Studi di Milano, 20122 Milan, Italy; 10036690@studenti.uniupo.it (S.C.); pierpaolo.biondetti@gmail.com (P.B.); 3Radiodiagnostic and Interventional Radiology Department, University of Eastern Piedmont, 28100 Novara, Italy

**Keywords:** interventional radiology, aneurysm, hemorrhage, subarachnoid, device, Woven EndoBridge, endovascular treatment

## Abstract

Introduction: The Woven EndoBridge (WEB) device is a self-expanding intrasaccular braided-wire device for the treatment of wide-neck bifurcation aneurysms (WNBAs). Even though this device has an excellent safety profile and a low risk of rebleeding, little is known about its long-term effects. Material and Methods: All patients treated with WEB due to ruptured WNBAs were subjected to follow-up digital subtraction angiography (DSA) at 2 and 3 years after device deployment. The degree of residual neck was assessed through BOSS, Lubicz, and WEBCAST scales. Data on modified Rankin scale (mRS), bleeding events, and ischemic events occurring during this time period were collected as well. Lastly, overall and procedure-related mortality rates were calculated. Results: A total of 21 patients were treated between 1 January 2016, and 31 December 2018. DSA demonstrated a patency grade of 57.1% and 61.1% at 2 and 3 years, respectively. The overall 2-year mortality rate due to causes unrelated to the aneurysm was 14.3%. None of the patients were retreated between the 2- and the 3-year follow-up. No rebleeding or stroke events occurred during the follow-up. Conclusions: WEB-treated ruptured aneurysms showed an excellent degree of stability over time. The overall mortality rate—unrelated to the procedure–observed in our sample was higher than what reported in the literature, a possible bias associated with the COVID-19 pandemic.

## 1. Introduction

Aneurysmal subarachnoid hemorrhage has an incidence of about 6.9/100,000 person-years in North America. Clipping, flow diversion, stenting, and coiling embolization all represent different treatment modalities for aneurysm exclusion [1].

Woven EndoBridge (MicroVention, Aliso Viejo, CA, USA) is a wire-interwoven nitinol (WIN) device designed for intrasaccular treatment of wide-neck bifurcation aneurysms (WNBAs) [2]. The increased number of wires in each single intertwined layer, along with their multiple diameters, allows WEB to obtain a rapid stasis of the contrast as well as a return of the radial force with deformation. This uniform response is clinically important for both small and large WEBs. To achieve successful aneurysm treatment, once deployed, the device has to acquire a stable intra-aneurysmal position and simultaneously remain flexible enough to conform safely to the sac. WIN ensures a radial force greater than that elicited by the arterial flow and, at the same time, lower than that of the compression exerted by the subarachnoid space on the aneurysm.

The WEBCAST, French Observatory, WEBCAST 2, and WEB-IT clinical trials, alongside several other studies, have retrospectively and prospectively analyzed the efficacy, safety, and feasibility of WEB [3,4,5,6]. In particular, The WEB-IT trial assessed 148 WEB-treated WNBAs, reporting complete occlusion or adequate occlusion in 53.8% and 84.6% of cases, respectively, during the 12-month follow-up. These measurements were made using the WEB-IT angiographic scale [4].

The ability of WEB to prevent rebleeding has been investigated by the CLARYS study [7]. Specifically, this trial included 56 ruptured WNBAs, showing a zero rate of rebleeding at 1 month and 1 year, indicating that an implanted WEB device is highly effective in preventing early and mid-late rebleeding.

Although there have been a growing number of studies on the safety and effectiveness of WEB in treating WNBAs, there is paucity of studies on the long-term effects of WEB performance in these patients. One of the few publications available in the literature is one by Pierot et al. 2021, assessing a 3-year follow-up of a small proportion of participants in the WEBCAST and WEBCAST-2 trials, reporting no delayed complications, a morbidity of 1.3%, and a mortality of 6.3% [8]. In this study, the angiographic analysis was only performed in 26.2% of the patients treated (16/61).

## 2. Materials and Methods

### 2.1. Study Design

The retrospective analysis was conducted in a single Italian center. Ethical review and approval were not required for this study in accordance with the national guidelines and institutional requirements.

### 2.2. Patient Population

All patients presenting a ruptured WNBA and treated with WEB devices between 1 January 2016, and 31 December 2018, were enrolled in the study. As inclusion criteria, we considered only patients with a long follow-up (>20 months) performed by digital subtraction angiography (DSA). The treatment indication for each patient was provided by a neurosurgeon and an interventional neuroradiologist multidisciplinary team. Demographics data were collected for statistical analysis.

### 2.3. Clinical and Radiological Assessments

Modified Rankin scale (mRS) at baseline and follow-up (2 years and 3 years), neurological adverse event (pre- and post-operative at 1 month after treatment and at 3-year follow-up) and morbidity data were collected. As “neurological adverse event”, we included all neurological events such as strokes and rebleeding.

Morbidity was defined as reported by Pierot et al. [6]: “An mRS score >2 when the preoperative mRS was ≤2 (or in case of ruptured aneurysm) and as an increase of 1 point when the preoperative mRS was >2”.

On first DSA, the WNBA location, 3D dimensions, and aneurysm neck were collected. On follow-up DSA after the embolization (at 2 and 3 years), the eventual neck residual was evaluated with Beaujon Occlusion Scale Score (BOSS), Lubicz scale, and the WEBCAST scale 9 as follows: “complete occlusions”, BOSS grade 0 or 0′, Lubicz grade A or B, and WEBCAST grade 1; “adequate occlusions”, BOSS grade 1, Lubicz grade C, and WEBCAST grade 2. “Aneurysm remnants”, BOSS grade 2, Lubicz grade D and WEBCAST grade 3. Table 1 summarize the principal characteristic of these scales.

### 2.4. Statistical Analysis

Descriptive analysis was performed for baseline demographics, aneurysm characteristics, and patient outcome metrics. Results are presented as frequencies and means with SD. All statistical analyses were performed with STATA 13 (version 13, StataCorp LLC College Station, TX, USA, Online Source, www.stata.com (accessed on 7 March 2022)).

## 3. Results

Thirty-five ruptured WNBAs were treated between 1 January 2016, and 31 December 2018. A total of 14 WBNAs (14 patients) were excluded from the study: eleven WBNAs did not perform a DSA follow-up, while one patient was lost during the follow-up. Thus, the main sample consisted of 21 patients whose demographic characteristics and WBNA localizations are shown in Table 2. 

The mean follow-up was 33.8 months (SD ± 5.03), about 3 years. At 3-year follow-up, we reported no aneurysm-related deaths (0.0%) and 3 non-aneurysm related deaths (14.3%, mean age: 81.3 months years, SD ± 3.98). Deaths occurred after a mean follow-up period of 29.0 months (SD ± 3.32). Deaths occurred in 2/3 patients from unspecified pulmonary causes and 1/3 from cardiovascular decompensation. No neurological adverse event (stroke events and rebleeding) or mRS worsening was reported at 3-year follow-up.

### Angiographic Results

The DSA imaging was acquired in 21/21 patients at 2-years (mean age 66.3 months, SD ± 10.75) and in 18/18 patients at 3 years (mean age 63.1 months, SD ± 8.13). DSA imaging analysis revealed complete occlusions in 12/21 (57.1%) WNBAs at 2-year follow-up and in 11/18 (61.1%) at 3-year follow-up. None of the patients had been retreated between the 2- and 3-year follow-ups (Table 3, Table 4 and Table 5).

No statistical differences were reported in the period between the two follow-ups. Analysis of follow-up DSA showed changes in WEB dispositive shape (compaction/compression) in 6/21 (28.6%) at 2 years and in 9/18 (50%) at 3 years.

## 4. Discussion

Since their FDA approval, WEBs have been increasingly adopted by many angiographic suites. Indeed, their unique structure allows treating acute cases of ruptured aneurysms without the use of antiplatelet and anticoagulant therapy. 

Although there are several studies on elective and emergency treatment, only a few of them have taken into account follow-ups beyond 2 years, and often these evaluations have been performed with contrast CT, MRI, or, even more rarely, DSA. Among them, the WEBCAST 2 trial and the study of Algin et al. evaluated WEB performance rispectively at 3-year follow-up and 7-years follow-up [8,9].

This monocenter and retrospective study aimed to describe the DSA findings in a cohort of WEB-treated WNBA patient at 2- and 3-year follow-up. Overall, our findings underscore the excellent stability of WEB-treated aneurysms, reporting a 0.0% of rebleeding. 

At both follow-ups, the morbidity was 0.0%: no neurological signs appeared in WEB-treated patients. These results are in line with previous studies, albeit the latter focused on relatively shorter follow-up periods [3,10,11], reporting a morbidity rate ranging from 0.0% to 12% at 1-year follow-up. 

Basilar apex aneurysms were the most frequently treated by WEB (57.1%), whereas middle cerebral artery bifurcation WBNAs were the least treated (4.8%). Our results are in contrast with previous studies showing that WBNAs of the middle cerebral artery bifurcation are the most frequently treated aneurysms with WEB—i.e., 48% in the Harker et al. meta-analysis [12]. This discrepancy may be due to the multidisciplinary approach adopted in our study, where a team formed by neurosurgeons and interventional neuroradiologists more often select basilar apex WBNA patients for endovascular treatment, while the middle cerebral artery WBNA were selected for neurosurgical clipping. This type of therapeutic choice was based on the results of current literature; particularly, Darsaut et al. and Smith et al. showed a better outcome of patients treated with surgical clipping compared with the endovascular approach for middle cerebral artery aneurysms [13,14].

Although incomplete occlusion was identified in 11.1% of cases—slightly higher than what reported by WEBCAST 2—, our findings attest an excellent stability of WEB over time. At both follow-ups, the reperfused neck neither had grown nor had it displayed a minimal increase in size (<2 mm). Hence, retreatment was not required (Figure 1).

According to the WEBCAST scale, the worsening rate at follow-up was 0.0%. In contrast, when we used the grading system of the Lubicz and BOSS scales, we observed worsening scores in 3/18 patients at 3-years—from A to B and from 0 to 0′, respectively. 

The analysis of the radiograms showed a change in the shape of the WEB device: at the control by DSA at 3 years, in 9 out of 18 patients (50%) the WEB changed shape; of these in 3/9 (33.3%) had a change in the BOSS scale.

The phenomenon seems to be time-related, with a shape deformation of 19.1% after 1-year from implantation [15]. This is consistent with the fact that WEBs tend to deform over time, taking on a concave aspect defined as “butterfly shape” by Janot et al. [15]. As a result, the neck partially retracts and the contrast medium leaks into the pouch, without WNBA sac reperfusion. However, according to Rosskopf et al. and Nawka et al., the shape deformation does not seem to influence WBNA sac occlusion [16,17]. In contrast to the study by Alig et colleagues in which shape modification was found in 45% of aneurysmal remnants, “butterfly shape” did not result in aneurysm reperfusions in our study [9].

The finding could support the thesis of Pierot et al. that the modification of shape could be indicative of a healing process of the aneurysm [18].

The excellent WEB stability may be ascribable to its ability to be endothelialized, a phenomenon recently described by Ding and co-workers [19]. This study assessed the stability of the device at 12 months in a group of WEB-implanted rabbits, reporting that 67% of aneurysms showed a complete neointima covered by endothelial cells across the aneurysm neck, while the sacs were completely filled with connective tissue and small unorganized thrombus.

In our long-tern follow-up sample, DSA imaging examination showed no worsening in terms of WEBCAST scores. Fittingly, after placement of the WEB device, there was no new reperfusion of the aneurysms sac after 2- and 3-years. One of the largest meta-analysis of coiling cerebral aneurysm treatment documented a reperfusion rate ranging between 7.2% and 29.6%—mean follow-up ranging from 4.7 to 38 months [12]. Comparing to the data coiling with the limitations of this study, WEB treatment would appear to have a lower reperfusion rate in this study.

Our sample has a mean age older than that observed by the previous studies—the mean age in the last WEB meta-analysis was 53.2 years, SD ± 15.8).

In addition, we show overall mortality to be higher than that reported by others (range between 1% and 10%) [7,20]. Particularly, Pierot et al. in 2021 reported a 6.3% of global mortality at 3 years, with 1 death related to the procedure and 4 deaths unrelated to the procedure. This bias might explain the difference in mortality: while the mortality rate for procedural complication was in the known range, the global mortality was uncharacteristically high. In this regard, the COVID-19 pandemic may have played a central role in affecting patient outcome. Of our sample, 2 of 3 deaths occurred from pulmonary causes during the pandemic period, suspected COVID-19.

WEB meta-analyses have pointed to a high rate of selection bias in the WEB studies analyzed [20,21,22]. In the study period, our Center treated all WNBA patients with WEB, adding quality to this long-term follow-up study [21,22]. Moreover, this study is the only one assessing a long-term follow-up exclusively by DSA, the gold standard technique for cerebral aneurysms.

One limitation of our study is that the sample was limited and collected in a single Center. Another limitation is that no comparison of randomized techniques has been performed.

## 5. Conclusions

Ruptured WNBAs treated with WEB showed a high stability at 2- and 3-years after implantation. Treated patients did not present increased morbidity at both follow-ups. However, the overall mortality was higher compared to that reported by the literature. In our limited sample, the rate of complete occlusion was 57.1% at 2 years and 61.1% at 3 years, and aneurysm retreatment was not required.

## Figures and Tables

**Figure 1 jcm-11-02879-f001:**
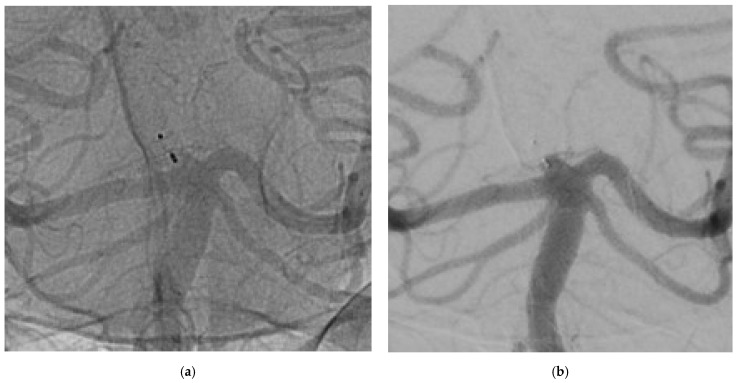
The images show a wide neck aneurysm of the apex of the basilar artery treated using WEB 3 × 2 mm. (**a**) not-substracted DSA at 3-years, (**b**) substracted DSA at 3-years. An aneurysm adeguate occlusion was detected and confirmed by 3D protocol.

**Table 1 jcm-11-02879-t001:** Describes the different scales used in the study: the Beaujon Occlusion Scale Score (BOSS), the Lubicz Scale, and the scale used in the WebCast study (WEBCAST Scale).

Beaujon Oc-Clusion Scale Score (BOSS)		Lubicz Scale		Webcast Scale	
0	No residual flowinside the aneurysm or the WEB	A	Complete occlusion, there is no contrast inside the aneurysm.	Grade 1	Complete occlusions
0′	Opacification of the proximal recess of the WEB	B	Complete occlusion with opacification of the proximal recess, there is no contrast inside the aneurysm but contrast fills in the central area below the WEB		
1	Residual flow inside the WEB	C	Neck remnant, there is contrast at the aneurysmal wall but no contrast in the aneurysm	Grade 2	Adequate occlusions
2	Neck remnant				
3	Aneurysm remnant	D	Aneurysm remnant, there is contrast in the neck and in the aneurysm or WEB	Grade 3	Aneurysm remnants
1 + 3	Contrast media depicted inside and around the device				

**Table 2 jcm-11-02879-t002:** Aneurysm occlusion degree in relation to its position at 2-year follow-up.

	2-Years	3-Years
Age	67.8 SD 10.83	63.1 SD 8.1
	N (%)	N (%)
TOT	21 (100%)	18 (100%)
Male	5 (23.8%)	3 (16.7%)
Female	18 (85.7%)	15 (83.3%)
Death	3 (14.3%)	0 (0.0%)
Aneurysm characteristics		
Neck	4.0 SD 1.12	4.0 SD 1.13
Aneurysm (h)	5.7 SD 1.66	6.2 SD 1.54
Aneurysm (l)	5.7 SD 1.55	6.3 SD 1.64
ICA	4 (19.0%)	4 (22.2%)
MCA	1 (4.8%)	1 (5.6%)
Acom	4 (19.0%)	2 (11.1%)
BA	12 (57.1%)	11 (61.1%)

ICA: internal carotid artery; MCA: middle cerebral artery; Acom: communicating artery; BA: basilar artery.

**Table 3 jcm-11-02879-t003:** Aneurysm occlusion degree in relation to its position at 2-year follow-up.

	ICA n = 4	MCA n = 1	Acom n = 4	BA n = 12
Complete occlusion	3	1	2	6
Adequate occlusion	1	0	2	6
Aneurysm remnant	0	0	0	0

ICA: internal carotid artery; MCA: middle cerebral artery; Acom: communicating artery; BA: basilar artery.

**Table 4 jcm-11-02879-t004:** Aneurysm occlusion degree in relation to its position at 3-year follow-up.

	ICA n = 4	MCA n = 1	Acom n = 2	BA n = 11
Complete occlusion	3	1	1	6
Adequate occlusion	1	0	1	5
Aneurysm remnant	0	0	0	0

ICA: internal carotid artery; MCA: middle cerebral artery; Acom: communicating artery; BA: basilar artery.

**Table 5 jcm-11-02879-t005:** WBNA occlusions according to different WEB scales, mortality rate, and neurological adverse effects at 2-year follow-up.

ID	Follow-Up	BOSS	Lubicz	WEB CAST	Death *	Rebleeding	Stoke	Neuro
1	24	0	A	1	-	-	-	-
2	22	2	C	2	-	-	-	-
3	23	3	D	3	-	-	-	-
4	22	0′	B	1	-	-	-	-
5	22	2	C	2	28	-	-	-
6	22	2	C	2	-	-	-	-
7	25	0′	B	1	-	-	-	-
8	26	0′	B	1	-	-	-	-
9	26	1	D	3	-	-	-	-
10	24	0′	B	1	-	-	-	-
11	22	2	C	2	32	-	-	-
12	23	0′	B	1	-	-	-	-
13	23	2	C	2	-	-	-	-
14	26	0′	B	1	-	-	-	-
15	22	0′	B	1	-	-	-	-
16	24	2	C	2	-	-	-	-
17	26	0	A	1	-	-	-	-
18	24	0′	B	1	-	-	-	-
19	22	2	C	2	-	-	-	-
20	26	2	C	2	-	-	-	-
21	20	0′	B	1	24	-	-	-

* Months from treatment to death.

## Data Availability

Not applicable.

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
