# Peer review of "Woven EndoBridge in Wide-Neck Bifurcation Aneurysms: Digital Subtraction Angiography at 3-Year Follow-Up"

_jcm, 2022, doi:10.3390/jcm11102879_

Round 1

Reviewer 1 Report

I read with interest the paper of Guzzardi et el. The authors present a long term follow up with DSA of 21 patients with wide-neck bifurcation aneurysms treated with The Woven EndoBridge (WEB) device with no patient needed retreatment and  patency grade of 57.1% and 61.1% at 2 and 3 years, evaluated by DSA. The results are consistent with the literature, however the mortality rate, although unrelated to the aneurysm,  in the series is higher than reported in the literature (14.3% vs range between 1% and 10% in the literature). As the authors point out this discrepancy may be due to the recent COVID pandemic, but the authors does not provide more specific details. If possible the authors have to provide more data on the cause of death for these patients.

In this series the Basilar apex aneurysms were the most frequently treated by WEB (57.1%), whereas middle cerebral artery bifurcation WBNAs were the least treated (4.8%), which is in contrast with the published literature where  WEB is used mostly in MCA aneurysm.  I think that this detail is particularly important as it reflects the true interdisciplinary approach the authors have in the choice of treatment, because according to the manuscript most of the MCA aneurysm are clipped. This correspond to our approach that we use in our clinic as well as data from the literature for better results of clipping in this type of aneurysms(1, 2)

I suggest that the authors create a separate table presenting the scales that they use to evaluate the eventual neck residual with DSA -Beaujon Occlusion Scale Score (BOSS), Lubicz scale, and the WEBCAST scale. I believe this will be useful because it will facilitate the understanding of Table 3, which now is somehow difficult to comprehend. 

References:

1.    Darsaut, T.E., et al., Surgical or Endovascular Management of Middle Cerebral Artery Aneurysms: A Randomized Comparison. World Neurosurg, 2021. 149: p. e521-e534.
2.    Smith, T.R., et al., Comparison of the Efficacy and Safety of Endovascular Coiling Versus Microsurgical Clipping for Unruptured Middle Cerebral Artery Aneurysms: A Systematic Review and Meta-Analysis. World Neurosurg, 2015. 84(4): p. 942-53.

Author Response

I formally thank the Reviewer 1 for reviewing and enriching the study.

Point 1. The authors present a long-term follow-up with DSA of 21 patients with wide-neck bifurcation aneurysms treated with The Woven EndoBridge (WEB) device with no patient needed retreatment and patency grade of 57.1% and 61.1% at 2 and 3 years, evaluated by DSA. The results are consistent with the literature, however the mortality rate, although unrelated to the aneurysm,  in the series is higher than reported in the literature (14.3% vs range between 1% and 10% in the literature). As the authors point out this discrepancy may be due to the recent COVID pandemic, but the authors does not provide more specific details. If possible the authors have to provide more data on the cause of death for these patients.

Response 1. I have added more details regarding the cause of death of the patients.

Point 2. In this series the Basilar apex aneurysms were the most frequently treated by WEB (57.1%), whereas middle cerebral artery bifurcation WBNAs were the least treated (4.8%), which is in contrast with the published literature where  WEB is used mostly in MCA aneurysm.  I think that this detail is particularly important as it reflects the true interdisciplinary approach the authors have in the choice of treatment, because according to the manuscript most of the MCA aneurysm are clipped. This correspond to our approach that we use in our clinic as well as data from the literature for better results of clipping in this type of aneurysms(1, 2)

Response 2. Thank you. I have added a comment to the study citing the interesting articles offered.

Point 3. I suggest that the authors create a separate table presenting the scales that they use to evaluate the eventual neck residual with DSA -Beaujon Occlusion Scale Score (BOSS), Lubicz scale, and the WEBCAST scale. I believe this will be useful because it will facilitate the understanding of Table 3, which now is somehow difficult to comprehend. 

Response 3: I added a new table, Table 1, for the used scales.

Minor errors and typos were checked and corrected.

Reviewer 2 Report

  Aneurysm occlusion degrees in relation to their position at the last follow-up should be added to the manuscript.

More comprehensive analyses and discussions about butterfly shape or WEB compaction/compression should be made. Please see and discuss more recent articles about these subjects (e.g., https://pubmed.ncbi.nlm.nih.gov/35098767/).

Author Response

I formally thank the Reviewer 2 for the valuable review performed on my article.

Point 1: Aneurysm occlusion degrees in relation to their position at the last follow-up should be added to the manuscript.

Response 1: I added a new table with this data called “table 4”

Point 2: More comprehensive analyses and discussions about butterfly shape or WEB compaction/compression should be made. Please see and discuss more recent articles about these subjects (e.g., https://pubmed.ncbi.nlm.nih.gov/35098767/).

Response 2: thanks for the advice, the article and bibliography have been updated with the cited article and an additional article from May 2022 by Nawka et al.

Round 2

Reviewer 2 Report

I recommend another review from my experienced colleague (Jonathan Cortese).

jonathan.cortese@aphp.fr